# Comparing eDNA metabarcoding primers for assessing fish communities in a biodiverse estuary

**Girish Kumar** [ID]*, **Ashley M. Reaume, Emily Farrell, Michelle R. Gaither***

Department of Biology, University of Central Florida, Genomics and Bioinformatics Cluster, Orlando, FL, United States of America

* girishkumar.nio@gmail.com (GK); michelle.gaither@ucf.edu (MRG)

**Data Availability Statement:** Raw demultiplexed fastq files have been uploaded to GenBank's Sequence Read Archive (SRA; BioProject ID: PRJNA767409) and the associated metadata can be found at the Genomic Observatories

## Abstract

Metabarcoding of environmental DNA is increasingly used for biodiversity assessments in aquatic communities. The efficiency and outcome of these efforts are dependent upon either *de novo* primer design or selecting an appropriate primer set from the dozens that have already been published. Unfortunately, there is a lack of studies that have directly compared the efficacy of different metabarcoding primers in marine and estuarine systems. Here we evaluate five commonly used primer sets designed to amplify rRNA barcoding genes in fishes and compare their performance using water samples collected from estuarine sites in the highly biodiverse Indian River Lagoon in Florida. Three of the five primer sets amplify a portion of the mitochondrial 12S gene (MiFish_12S, 171bp; Riaz_12S, 106 bp; Valentini_12S, 63 bp), one amplifies 219 bp of the mitochondrial 16S gene (Berry_16S), and the other amplifies 271 bp of the nuclear 18S gene (MacDonald_18S). The vast majority of the metabarcoding reads (> 99%) generated using the 18S primer set assigned to non-target (non-fish) taxa and therefore this primer set was omitted from most analyses. Using a conservative 99% similarity threshold for species level assignments, we detected a comparable number of species (55 and 49, respectively) and similarly high Shannon's diversity values for the Riaz_12S and Berry_16S primer sets. Meanwhile, just 34 and 32 species were detected using the MiFish_12S and Valentini_12S primer sets, respectively. We were able to amplify both bony and cartilaginous fishes using the four primer sets with the vast majority of reads (>99%) assigned to the former. We detected the greatest number of elasmobranchs (six species) with the Riaz_12S primer set suggesting that it may be a suitable candidate set for the detection of sharks and rays. Of the total 76 fish species that were identified across all datasets, the combined three 12S primer sets detected 85.5% (65 species) while the combination of the Riaz_12S and Berry_16S primers detected 93.4% (71 species). These results highlight the importance of employing multiple primer sets as well as using primers that target different genomic regions. Moreover, our results suggest that the widely adopted MiFish_12S primers may not be the best choice, rather we found that the Riaz_12S primer set was the most effective for eDNA-based fish surveys in our system.

MetaDatabase (GEOME; https://geome-db.org/;
(89)). ASV files with read counts and
corresponding sequences can be found on Data
Dryad (DOI: https://doi.org/10.5061/dryad.
70rxwdbzc).

**Funding:** This work was made possible with
funding from the University of Central Florida and
the University of Florida Sea Grant Program to M.
R.G, from the Preeminent Postdoctoral Program at
the University of Central Florida to G.K., from the
Trustee's Fellowship at the University of Central
Florida and the Florida Fish and Wildlife
Conservation Commission Forage Fish Fellowship
to E.F., and from the NOAA Margaret A. Davidson
Graduate Fellowship at the National Estuarine
Research Reserves to A.R. and M.R.G.The funders
had no role in study design, data collection and
analysis, decision to publish, or preparation of the
manuscript.

**Competing interests:** The authors have declared
that no competing interests exist.

# 1 Introduction

The monitoring of marine and aquatic communities requires accurate biodiversity assessments which are typically based on surveys conducted using nets, traps, cameras, or direct observation. Environmental DNA (eDNA) approaches are emerging as a tool for the characterization of marine biodiversity that can complement traditional surveys [1–4]. The initial eDNA studies in aquatic systems were published a decade ago and focused on species-specific primers and qPCR to detect invasive or endangered freshwater species [5–7]. Not long thereafter, researchers began exploring the use of eDNA in marine environments [8] including the use of metabarcoding approaches to identify and characterize whole communities [9–11]. This latter approach involves PCR amplification and library preparation using primers that are designed to amplify a barcoding gene (i.e., COI, 16S, 18S) across a specific taxonomic group; the breadth of which can vary from metazoans to a single genus. Subsequent sequencing of the metabarcoding libraries on high-throughput sequencing platforms such as an Illumina MiSeq generates millions of sequence reads that must then be sorted bioinformatically and assigned a specific taxonomy.

Prior to beginning a metabarcoding initiative, primers that target the group of interest must be selected from the literature or designed *de novo*. While there are a number of published primer sets already available [12], each suffers from unique biases making primer testing essential. For instance, primer-template mismatch or unequal amplification efficiency (PCR bias) can result in uneven amplification and false negatives [13–15]. This is particularly true for primers designed to work across broad taxonomic groups where primer mismatches are likely to occur. PCR bias can also result from the competition of unbound primers for DNA templates, in which case the most abundant templates are more likely to be amplified, resulting in false negatives for low copy number templates [16,17]. Furthermore, if PCR conditions allow for non-specific binding (low annealing temperature) or if poorly designed primers permit for ubiquitous template binding, preferential amplification of the most common templates can lead to sequencing runs dominated by non-target taxa such as bacteria, fungi, or algae at the expense of less common target DNA.

Cytochrome *c* oxidase subunit I (COI) has historically been the most common gene for DNA barcoding in animals [18–21]. As a result, there is extensive reference data available for COI in the public databases (National Center for Biotechnology Information, NCBI; the Barcode of Life Data System, BOLD). However, the conserved nature of COI, which makes it useful as a barcoding gene, also complicates the design of taxon-specific primers for metabarcoding [22–24]. Consequently, COI metabarcoding primers are designed with a high degree of base degeneracy [25,26] which results in the amplification of a wide breadth of taxa [27], but also often results in a large percentage of non-target sequence reads [28,29]. As a result, COI is not commonly used for eDNA metabarcoding [but see 18,30]. Instead, the mitochondrial 12S and 16S rRNA genes are generally favored [16,31–36]. Both genes include conserved regions for primer design as well as variable regions that allow for genus or species-level resolution. Moreover, reference sequences for these two genes, particularly for fishes, are well represented in the public databases [23,37].

Before beginning an experiment, the performance of newly designed primers can be evaluated *in silico* [16], however, there is no guarantee that they will perform as predicted when applied to field-based samples. The efficiency of metabarcoding primers varies with community composition and complexity [16,22,38], and as a result, it is recommended that candidate primer sets be tested on field samples [29]. Over the past several years, a number of studies have been published that evaluate and compare different primer sets for use in eDNA metabarcoding of fishes [16,29,39–43]. In a recent study, Zhang et al. [42] evaluated the efficacy of 22

published metabarcoding primer sets covering several gene regions and using both *in silico* and *in vitro* analysis on freshwater fishes in China. They found that, in general, 12S rRNA primers detected a greater number of species than either 16S rRNA or COI primers. Moreover, they found inconsistent results when comparing *in silico* and *in vitro* experiments [42]. Here we focus on five primer sets (12S and 16S), three of which performed well in the freshwater study of Zhang et al. [42], and include an additional 18S primer set not previously evaluated (but designed for fishes). We assess their suitability for detecting fishes in a species rich estuarine system in Florida (the Indian River Lagoon) and compare them in terms of 1) specificity (amplification of fishes at the exclusion of other taxa), 2) universality (amplification of a diversity of fish taxa), and 3) taxonomic resolution (ability to resolve taxa to the species level).

## 2 Materials and methods

### 2.1 Primer selection and sample collection

Here we selected five primer sets to evaluate. These included three 12S rRNA and one 16S rRNA primer set (Table 1) that had been employed in a number of aquatic eDNA studies targeting fishes. In addition, we selected one nuclear 18S rRNA primer set [44] that was designed for freshwater fishes but had not been evaluated for eDNA metabarcoding.

Two replicate 500 ml water samples were collected in June 2018 from each of six sites within the northern Indian River Lagoon (IRL) in central Florida (S1 Table). Sites were in close proximity and over a similar shallow water habitat known to support oyster reefs. Water samples were taken at the surface. Prior to water sampling, all collection bottles, forceps, scissors, and filter holders were sterilized with 20% sodium hypochlorite solution for at least 20 min, rinsed with reverse osmosis (RO) water and air dried. Water samples were collected in sterilized Nalgene bottles including two negative field controls consisting of sterile Nalgene bottles filled in the field with store-bought bottled water. As is true for much of the IRL, our water samples were highly turbid with high concentrations of suspended organic and/or inorganic material [45]. Following the recommendations of Kumar et al. [46], samples were stored on ice and transported back to the lab and filtered using 0.45 μm mixed cellulose ester (MCE) filters. Negative laboratory controls consisted of 500 ml of RO water filtered using the same protocol as our field samples. After filtration, filter membranes were stored in 3 ml tubes in Longmire's buffer at -20°C until DNA extraction.

### 2.2 Library preparation and sequencing

All DNA extractions were carried out in a dedicated PCR free workspace at the University of Central Florida Marine Molecular Ecology and Evolution Laboratory. To prevent

**Table 1. Primer sets used in this study including primer sequence, annealing temperature used for PCR1, and expected average amplicon length.**

| Primer set | Locus | Original primer name | Primer sequence (5'-3') | Annealing temperature (°C) | Amplicon length (bp) | Reference |
|---|---|---|---|---|---|---|
| MiFish_12S | 12S | MiFish-U-F MiFish-U-R | GTCGGTAAAACTCGTGCCAGC CATAGTGGGGTATCTAATCCCAGTTTG | 61.5 | 171 | Miya et al. 2015 |
| Riaz_12S | 12S | 12S-V5f 12S-V5r | ACTGGGATTAGATACCCC TAGAACAGGCTCCTCTAG | 55 | 106 | Riaz et al. 2011 |
| Valentini_12S | 12S | L1848 H1913 | ACACCGCCCGTCACTCT CTTCCGGTACACTTACCATG | 55 | 63 | Valentini et al. 2016 |
| Berry_16S | 16S | Fish16sF/D 16s2R | GACCCTATGGAGCTTTAGAC CGCTGTTATCCCTADRGTAACT | 54 | 219 | Berry et al. 2017 |
| MacDonald_18S | 18S | Fish_18S_1F Fish_18S_3R | GAATCAGGGTTCGATTCC CAACTACGAGCTTTTTAACTGC | 62 | 271 | MacDonald et al. 2014 |

contamination, all equipment and bench spaces were cleaned before use with 10% sodium hypochlorite solution followed by 70% ethanol and irradiated with UV light for 20 min. All pipetting was conducted using sterile barrier filter tips. Prior to DNA extraction, filters were cut in half and one-half was archived in Longmire's buffer and stored at -20˚C. The other half was placed in a 1.5 ml Eppendorf tube for DNA extraction and cut into small pieces using sterilized scissors. DNA was extracted from each and eluted in 100 μL of buffer using the E.Z.N.A. Tissue DNA Kit (Omega Bio-tek, Inc., GA, USA), following the manufacturers' protocol and which has been shown to perform well in previous experiments [45]. The resulting extraction was purified using a Zymo OneStep PCR Inhibitor Removal Kit (Zymo Research, CA, USA), and eluted in a final volume of 50 μl. DNA concentrations were determined using a Qubit 4.0 and the dsDNA High Sensitivity Assay Kit (Invitrogen, CA, USA).

Illumina libraries were constructed for each sample using a two-step PCR protocol following Kumar et al. [45]. PCR 1 (qPCR) was performed using custom primers that included both Illumina sequencing primers and locus specific primers (see Table 1 in [45]). Amplifications were carried out on a CFX96 Touch Real Time PCR System (Bio-Rad, CA, USA) in a total volume of 25 μl with each reaction containing 12.5 μl of 2× SsoAdvance Universal SYBR Green Supermix (Bio-Rad), 0.5 μl forward primer (10 μM), 0.5 μl reverse primer (10 μM), 2 μl of template DNA, and 9.5 μl of ultrapure water (ThermoFisher Scientific, MA, USA). The thermocycling profile included an initial denaturation step at 95˚C for 3 min, followed by 30 cycles of denaturation at 95˚C for 30 s, annealing at the primer annealing temperature (Table 1) for 30 s, and extension at 72˚C for 30 s, followed by a final extension at 72˚C for 5 min. Each qPCR run included a no-template control as well as an extraction control. To minimize false negatives (PCR dropouts), qPCRs were performed in duplicates. Following qPCR, duplicates were pooled before excess primers and dNTPs were removed using an E.Z.N.A. Cycle Pure Kit (Omega Bio-tek, Inc.) following manufacturer's protocol. The purified PCR products were quantified using a Qubit 4.0 fluorometer and served as the template DNA for PCR 2.

PCR 2 was performed using primer pairs consisting of Illumina adaptors (P5 and P7), 8 bp Nextera index sequences, and an overhang sequence complementary to the Illumina sequencing primer (see S1 Fig in [45]). Amplifications were carried out using a Veriti Thermal Cycler (Applied Biosystems, CA, USA) and 25 μl reaction volumes containing 12.5 μl IBI Taq 2× Master Mix (IBI Scientific, IA, USA), 0.5 μl forward primer (10 μM), 0.5 μl reverse primer (10 μM), 2 μl DNA template, and 9.5 μl of ultrapure water. PCR cycling conditions were identical to PCR 1 except only 15 cycles were run and a universal annealing temperature of 55˚C was employed. Final PCR products were cleaned using E.Z.N.A. Cycle Pure Kits, quantified on a Qubit 4.0, and pooled in equimolar concentrations (one pool for each primer set). Each pooled library was then size-selected based on expected fragment size using a PippenHT (Sage Science, MA, USA) and a 2% agarose gel cassette and quantified using a NEB Next Library Quantification Kit for Illumina (New England Biolabs, MA, USA). The library was adjusted to 4 nM and denatured following Illumina protocols. The denatured library was combined with 10% PhiX control and sequenced bidirectionally on an Illumina MiSeq at the University of Central Florida Genomics and Bioinformatics Cluster (GBC) Core Laboratory. Sequencing was conducted using a Nano 300 v2 (2 × 150) Reagent Kit for 2×111 cycles and a Nano 500 v2 (2 × 250) Reagent Kit for 2×251 cycles, depending on amplicon size.

## 2.3 Bioinformatic processing

The Illumina sequencing data was demultiplexed using the Illumina MiSeq software and downloaded onto an in-house server maintained by the Genomics and Bioinformatics Cluster (GBC) at the University of Central Florida. Individual FASTQ files were then filtered following

a series of quality control steps using USEARCH v10 [47] and VSEARCH v2.14 [48]. First, the forward and reverse reads were merged using the fastq_mergpairs command in USEARCH with a minimum overlap of 100 bp for MiFish_12S, Berry_16S, MacDonald_18S; and 60 bp for Valentini_12S and Riaz_12S, and a maximum number of mismatches set at 3 bp. Sequences of unexpected length were discarded using the -fastq_minlen command of USEARCH to retain only those reads with maximum deviation of 10% from the minimum amplicon length. To locate and remove primers, the merged reads were sub-sampled to 5,000 sequences and then VSEARCH was used to remove primers. Next, we dereplicated sequences and discarded sequences with expected errors > 0.5 using VSEARCH. Finally, unique sequences were denoised using the UNOISE3 [49] option implemented in USEARCH. UNOISE3 generates zero radius operational taxonomic units (ZOTUs) by correcting point errors and filtering chimeric sequences [49]. To minimize the chance of spurious sequences being included in the final dataset, the minimum abundance of five reads were set to generate amplicon sequence variants (ASVs).

### 2.4 Taxonomic assignments

ASVs were compared against the NCBI GenBank nucleotide database using BLASTn with the default parameters. To retrieve the full taxonomic identity, we queried each of the "taxids" from the BLAST results against the NCBI database using the "taxonkit lineage" command in the program TaxonKit [50]. To reduce the uncertainty in taxonomic assignments, we discarded ASVs with a bitscore below 250 and/or query coverage below 100%. Each ASV was then assigned to the lowest taxonomic level based on the percent similarity to NCBI alignments. We recognize that the rate of evolution varies across genes and gene regions and so setting a single taxonomic threshold and applying it across all primer sets could impact interpretations of the results. Therefore, we examined results for three similarity thresholds for species level designations (99%, 98%, and 97%) and present these in supplemental materials (S2 Table). These results did not change our interpretation of the data and therefore in main manuscript we present the results for the following taxonomic thresholds for all primer sets: 99% for species; 97% for genus; 95% for family; 90% for order; 85% for class; and 80% for phylum following West et al. [35]. The resulting list of species was checked against a list of known species from the Indian River Lagoon. Species detected in our metabarcoding data but absent from that list were queried against FishBase (www.fishbase.org) to determine if they were present in the central-west Atlantic. If an ASV matched ≥ 99% with two closely related species but only one of them was known to occur in the Indian River Lagoon or eastern Atlantic, then that ASV was assigned to that known taxon. However, if both species were known to occur in the eastern Atlantic region, taxonomic assignments were collapsed to the genus level.

### 2.5 Statistical analyses

Unless otherwise specified, all statistical analyses were performed using R version 4.0.2 [51]. For data analyses, sequence reads from the two replicates taken at each of the six sampling sites were pooled. We computed diversity indices (species richness, evenness, and Shannon's diversity) using the BiodiversityR package v. 2.12–3 [52] and standard deviations represent variation across the six sample sites. We ran an analysis of variance (ANOVA) using the program JMP Pro 12 (SAS Institute Inc., NC, USA) to determine if diversity indices differed among sample sites and across primer sets. When a significant interaction was detected, we performed a post-hoc Tukey-Kramer Honest Significant Difference (HSD) test to determine which group means were significantly different. Dissimilarity in species composition among the different primer sets were calculated by non-metric multidimensional scaling (NMDS)

analysis using read abundance-based on Bray-Curtis coefficients. The NMDS analysis was conducted using metaMDS commands in the R package Vegan [53] and visualized in RStudio using ggplot2 [54]. An analysis of similarity (ANOSIM; [55]) was used to test for significance. When a significant difference was detected, we performed the similarity percentage analysis (SIMPER) in Vegan to determine which taxa were responsible for explaining most of the difference among groups. Finally, to visualize the number of common and unique species detected across primer sets, a Venn diagram was constructed using the R package VennDiagram v1.6.2 [56].

## 3 Results

### 3.1 Illumina sequencing

A total of 2.1 and 2.0 million paired-end sequence reads were generated from the $2 \times 150$ bp (Riaz_12S and Valentini_12S) and the $2 \times 250$ bp (MiFish_12S, Berry_16S, and MacDonald_18S) Illumina MiSeq runs, respectively. Across both runs ~ 92% of the paired-end reads had phred scores of $\geq 30$ and in total 63.19% of the reads were retained after quality filtering. The percentage of reads retained after quality control was 71.02% for MiFish_12S, 63.74% for Riaz_12S, 60.10% for Valentini_12S, 78.74% for Berry_16S, and 72.25% for MacDonald_12S with the number of total reads retained for each primer set ranging from 295,277 to 711,351 (Table 2). The average number of reads per sample was the highest for Berry_16S (57,729 ± 14,331 reads) and lowest for MiFish_12S (33,711 ± 13,984) while the Riaz_12S and Valentini_12S primers resulted in 51,434 ± 13,944 and 48,522 ± 13,511 reads, respectively.

There was no indication of contamination in any of the extraction or PCR negative controls (samples did not amplify in PCR 1). However, two out of the six field negative controls did amplify and so these were included in library preparation and sequencing. Sequencing these negative field controls resulted in just 5,206 reads, 99.23% (5,166 reads) of which assigned to human DNA for Valentini_12S primers. Of the remaining 40 reads, 10 were assigned to *Lutjanus spp.* and 28 reads were assigned to *Lutjanus griseus* for the Riaz_12S primers, while the other two reads were assigned to *Mugil curema* for the Valentini_12S primers. No contaminating sequences were detected using the other three primers sets (MiFish_12S, Berry_16S, MacDonald_18S).

The MacDonald_18S primer set amplified taxa from 24 different phyla including Porifera, Cnidaria, Platyhelminthes, Nematoda, Mollusca, Annelida, Arthropoda, Echinodermata, and Chordata. Copepods (phylum Arthropoda) accounted for 28.57% of reads while green algae (phylum Chlorophyta) accounted for 24.87% of reads. Since only 278 reads or 0.09% of the

**Table 2. Illumina sequencing results.**

| # Sequence reads | MiFish_12S | Riaz_12S | Valentini_12S | Berry_16S | MacDonald_18S |
|---|---|---|---|---|---|
| After quality filtering | 497024 | 637771 | 676856 | 711351 | 295277 |
| **% assigned to fish** | | | | | |
| Class | 94.12 | 99.99 | 94.49 | 99.95 | 0.094 |
| Order | 92.05 | 99.99 | 94.45 | 99.95 | 0.046 |
| Family | 84.88 | 98.31 | 88.51 | 98.11 | 0.00 |
| Genus | 84.88 | 98.31 | 88.51 | 98.11 | 0.00 |
| Species | 81.39 | 96.78 | 86.03 | 97.38 | 0.00 |
| Non-target | 5.88 | 0.005 | 4.75 | 0.055 | 99.99 |

Listed is the total number of reads retained after quality control and the percentage of those reads assigned to target (fishes) and non-target taxa (non-fish).

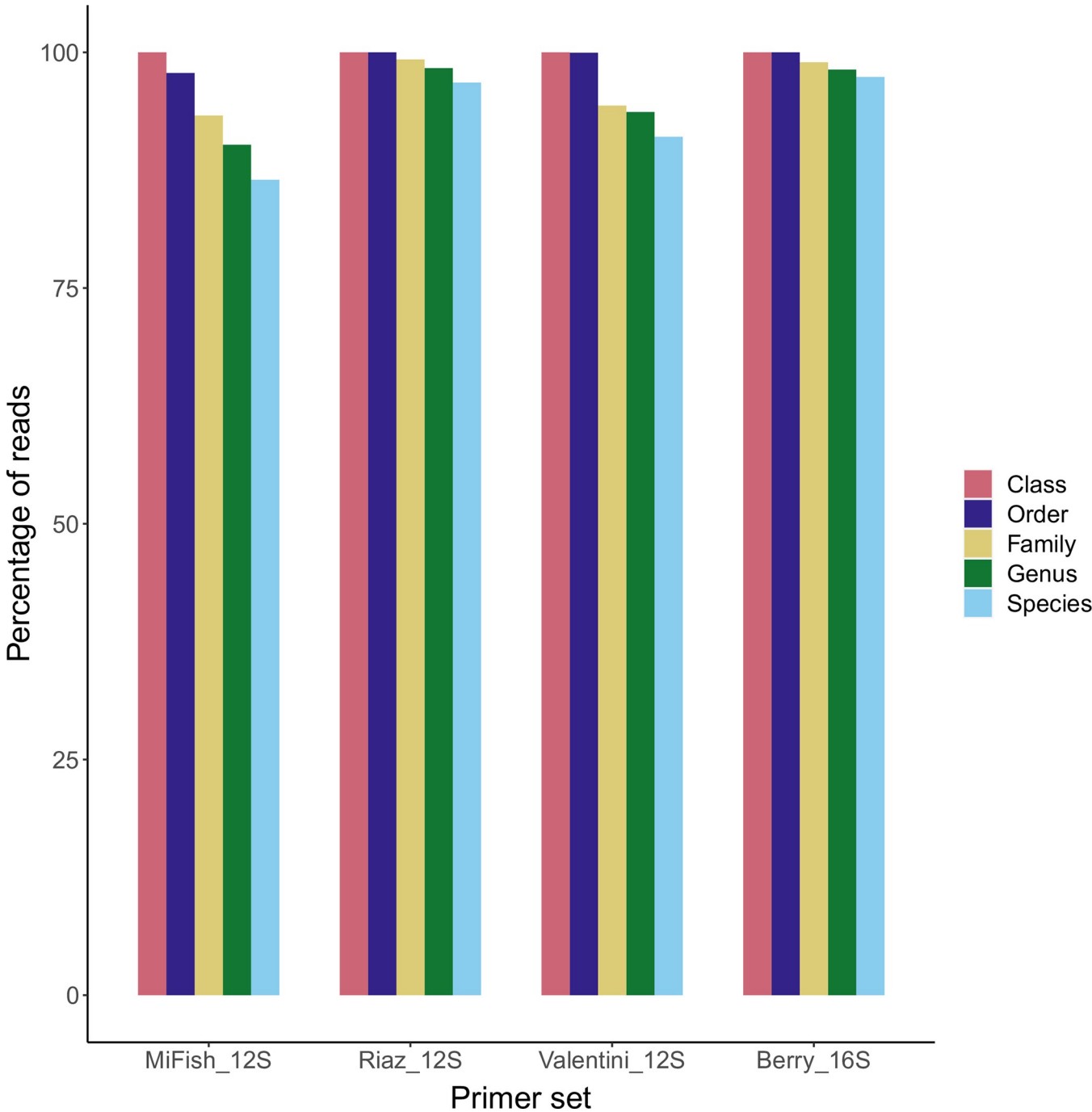

**Fig 1. Proportion of reads assigned to each taxonomic level for the four metabarcoding primer sets tested here.**

total reads that resulted from the MacDonald_18S primer set were assigned to fishes (and none could be resolved to species level), we did not include data of this marker in subsequent analyses. Across the remaining four primer sets, the proportion of reads passing quality filters that were assigned to fish taxa were 94.12% for MiFish_12S, 99.99% for Riaz_12S, 94.49% for Valentini_12S, and 99.95% for Berry_16S (Table 2, Fig 1).

**Table 3. Number of fish taxa detected for each of the four metabarcoding primer sets using the following similarity thresholds: 99% for species; 97% for genus; 95% for family; 90% for order; 85% for class; and 80% for phylum.**

| Taxonomic rank | MiFish_12S | Riaz_12S | Valentini_12S | Berry_16S | Total |
|---|---|---|---|---|---|
| Class | 2 | 2 | 2 | 2 | 2 |
| Order | 14 | 16 | 12 | 12 | 17 |
| Family | 28 | 37 | 24 | 29 | 48 |
| Genus | 38 | 53 | 34 | 43 | 67 |
| Species | 34 | 55 | 32 | 49 | 76 |
| ASVs | 161 | 159 | 247 | 140 | - |

Classes include Actinopterygii and Chondrichthyes.

## 3.2 Taxonomic assignments and biodiversity estimates

While adjusting the sequence similarity threshold did change the number of species detected for each primer set, the interpretation of the overall pattern did not change. Regardless of which similarity threshold was applied the Riaz_12S primer set resulted in the most species detected with Berry-16S ranking second (S2 Table). There were no notable changes in taxa detected when comparing the 97% and 98% thresholds but there was predictably an increase in the number of species resolved if we applied the 98% threshold compared to the more conservative 99% threshold. Most notably, when we compared the best performing Riaz_12S using the 99% cutoff with the worst performing primers sets using a 97% cutoff the latter still resulted in a higher number of species detections (S2 Table). Similar patterns were found for the diversity indices (S3 Table). As a result, here we apply a single conservative similarity threshold of 99% for species-level designations and report these throughout.

Across the four primer sets, we detected 76 species of fish in two classes, 17 orders, 48 families, and 67 genera (Table 3). Although the vast majority of reads were assigned to class Actinopterygii (>99%; bony fish), we also detected sharks and rays with each primer set (class Chondrichthyes; Table 4), but with relatively low read counts that ranged from 7 to 2,581 reads. We detected the highest number of taxa (across all taxonomic levels) with the Riaz_12S primer set but the highest number of ASVs with the Valentini_12S primers (Table 3). We resolved a similar number of fish species with the Riaz_12S and Berry_16S primer sets (55 ± 5.59 and 49 ± 6.69, respectively). The MiFish_12S and Valentini_12S primers sets also resolved similar numbers of species (34 ± 3.13 and 32 ± 4.72, respectively) but significantly fewer than the Riaz_12S and Berry_16S primer sets (Table 5; Tukey-Kramer HSD $P < 0.05$).

**Table 4. Species of cartilaginous fishes detected with each of the four metabarcoding primer sets tested in this study.**

| Class | Subclass | Primer | Species |
|---|---|---|---|
| Chondrichthyes | Elasmobranchii | MiFish_12S | *Dasyatis say* |
| | | Riaz_12S | *Aetobatus narinari* <br> *Dasyatis sabina* |
| | | | *Dasyatis say* |
| | | | *Gymnura micrura* <br> *Hypanus sabinus* <br> *Rhinoptera spp.* |
| | | Valentini_12S | *Aetobatus narinari* <br> *Dasyatis say* <br> *Hypanus americanus* <br> *Rhinoptera spp.* |
| | | Berry_16S | *Hypanus americanus* |

**Table 5. Diversity indices including species richness, evenness, and Shannon's diversity calculated using the R package BiodiversityR v. 2.12–3 for each of the four metabarcoding primer sets.**

| Primer set | Species richness | Evenness | Shannon's diversity |
|---|---|---|---|
| MiFish_12S | 34 ± 3.13 [a,c] | 0.134 ± 0.054 [a] | 1.01 ± 0.45 [a] |
| Riaz_12S | 55 ± 5.59 [b] | 0.163 ± 0.067 [a] | 1.75 ± 0.47 [b] |
| Valentini_12S | 32 ± 4.72 [a] | 0.153 ± 0.054 [a] | 1.09 ± 0.35 [a,b] |
| Berry_16S | 49 ± 6.69 [b,c] | 0.138 ± 0.064 [a] | 1.30 ± 0.52 [a,b] |

Standard deviations are based on the data from all six sample sites. Those comparisons that were significant using an ANOVA show superscripts with different letters indicating statistical difference at $P < 0.05$ using the post-hoc Tukey-Kramer Honest Significant Difference test.

Estimates of evenness were low and were significantly different among primer sets (Table 5). Our estimates of Shannon's diversity ranged from 1.01 ± 0.45 for MiFish_12S to 1.75 ± 0.47 for Riaz_12S. Similar to species richness, Shannon's diversity values were roughly equal for the MiFish_12S (1.01 ± 0.45) and Valentini_12S (1.09 ± 0.35) primer sets and were not significantly different (Table 5). However, Shannon's diversity values for the Riaz_12S primer set were significantly higher than the MiFish_12S primer set (Tukey-Kramer HSD, $P = 0.049$; Table 5).

Of the 76 species of fish detected in this study (S1 Fig), only 17 were common to all datasets with between 1–12 unique species detected by any single primer set (Fig 2). The numbers of reads assigned to the 17 common taxa were 514,041, 555,883, and 569,976 for the Riaz_12S, Valentini_12S, and Berry_16S primer sets, respectively (S4 Table). The MiFish_12S had the lowest number of reads across these 17 species (381,805) and accounted for just 16% of total read counts across all datasets (S5 Table) suggesting lower PCR efficiency. We resolved a high number of unique species using the Riaz_12S and Berry_16S primer sets (12 and 11 species, respectively), while only three unique species were detected using the MiFish_12S primers and only one was detected using the Valentini_12S primers. Based on the ANOSIM analyses, species assemblages differed significantly across primer sets ($R = 0.4011$, $P < 0.001$); a finding that was supported by the NMDS plots which showed clear separation among marker sets when either the read abundance data (Fig 3) or presence/absence data (S2 Fig) were analyzed. However, no significant differences were observed in the fish communities across the different sampling sites (ANOVA, $F = 1.25$, $P = 0.33$; Fig 3). The five most influential species contributing to these differences, based on SIMPER analyses, are given in Table 6 with the White mullet *Mugil curema* ranking first in all comparisons. This species was also the most dominant in terms of read count across primer sets.

## 4 Discussion

When designing eDNA studies, choosing metabarcoding primers is of critical importance as the initial monetary investment can be high and the decision will have a significant impact on project results [29,42]. Primers with insufficient taxonomic specificity can result in the loss of sequencing effort to non-target taxa as well as false negatives. This is particularly true in highly diverse study systems where non-target DNA (i.e., microbial and plankton communities) is abundant [29]. Despite the growing interest in the use of eDNA to assess fish communities, there has been a surprising lack of studies that have directly compared the efficacy of metabarcoding primer sets in marine and estuarine systems. Because the performance of eDNA metabarcoding primers will vary depending on the study system (freshwater, marine, or estuarine) and taxonomic composition, there is no guarantee that a primer set that performs well in a freshwater system will do so in marine or estuarine systems. For this reason, we included three

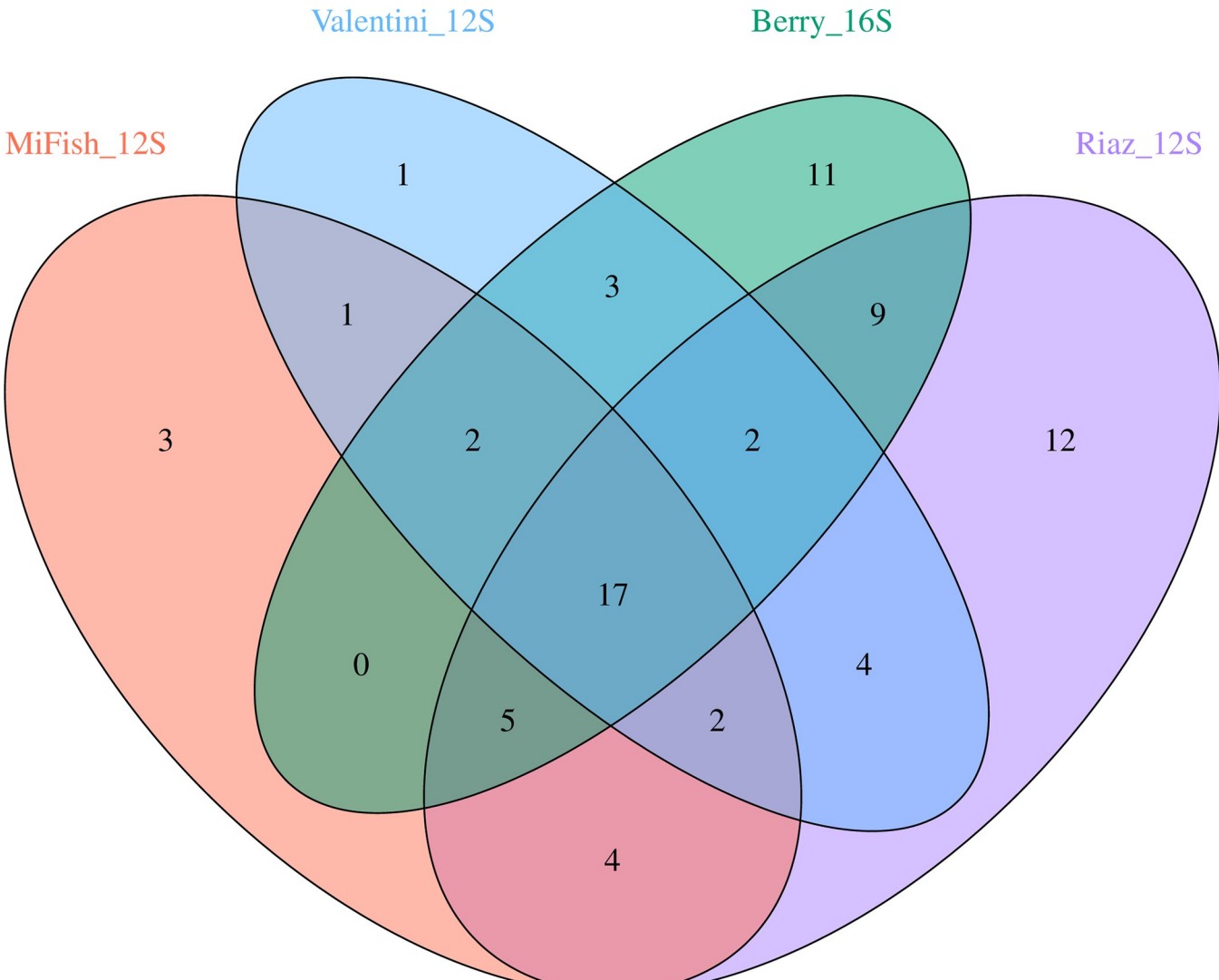

**Fig 2. Venn diagram representing the number of fish species detected across four metabarcoding primer sets.** The numbers shown in the areas of overlap reflect shared species.

primer sets that performed well in the freshwater study of Zhang et al. [42], to determine their effectiveness in a biodiverse estuarine system. Our study system, the Indian River Lagoon, which runs along Florida's east coast is regarded as the most species rich estuary in the U.S. [57]. Our results show that the 12S rRNA primers of Riaz et al. [58] were the most taxon-specific with 99.99% of the resulting reads assigned to fish and 96.78% of reads assigned to the species-level. The 16S rRNA primers of Berry et al. [31] also performed well with 99.95% of reads assigned to fish and 97.38% assigned to species. Furthermore, a similar number of species were identified using these primer sets (Table 3) and both resulted in comparatively high Shannon's diversity values (Table 5). The popular 12S primers of Miya et al. [59] did well in terms of the percent reads assigned to fish taxa (94.12%) and species (81.29%), but only 34 species were detected using this primer set, performing similarly to the Valentini_12S primer set. Taken together, these results indicate that the Riaz_12S and Berry_16S primer sets performed best in our biodiverse estuarine system and resolved the greatest number of target species. Moreover, of the 76 fish species identified in this study, 85.5% (65 of 76) were detected when

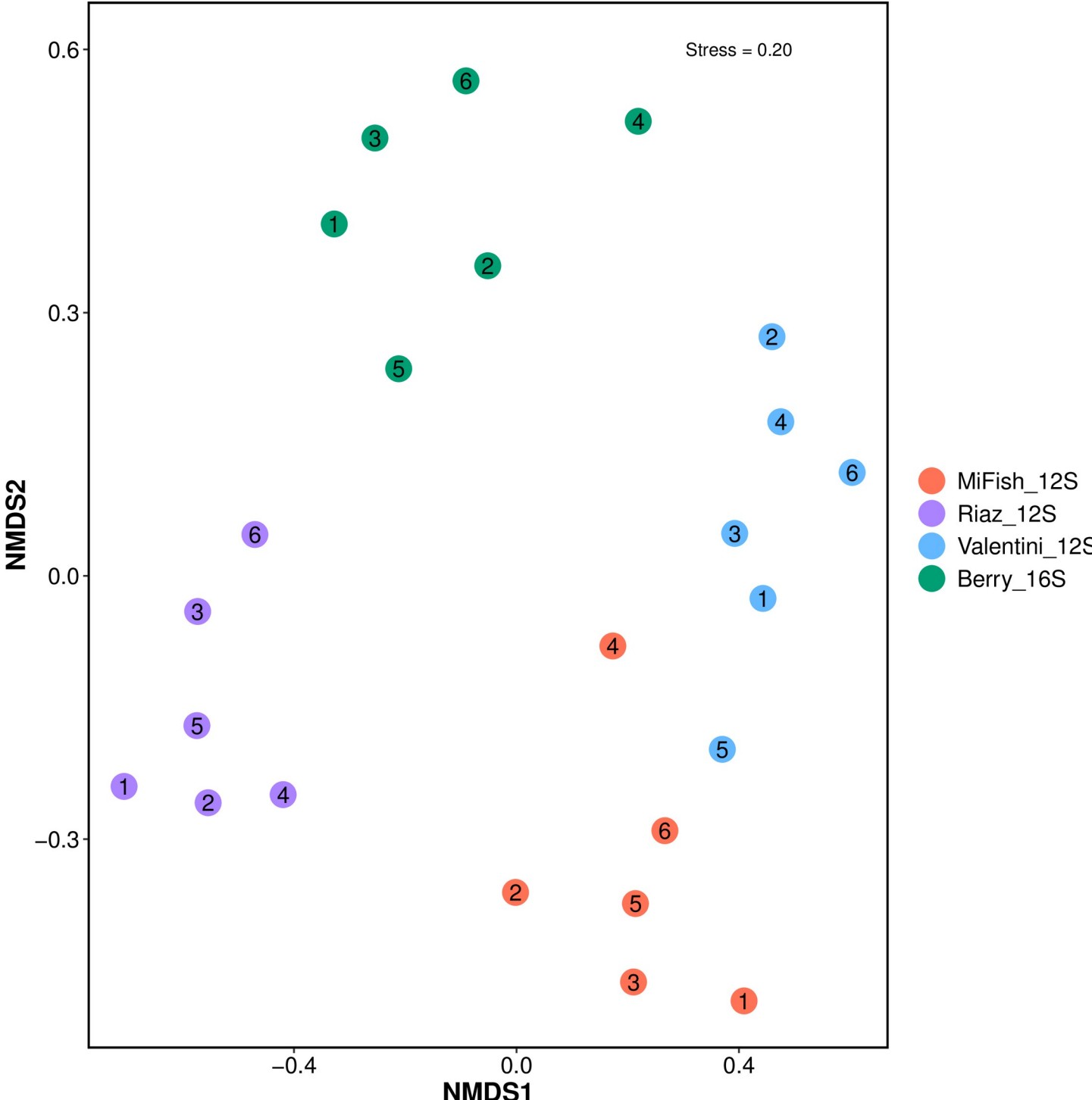

**Fig 3. Nonmetric multidimensional scaling plots (NMDS) based on read abundance for each of the species detected from six sampling locations in the Indian River Lagoon, Florida.** The numbers inside the circles represent sample sites. Metabarcoding data was generated for four primer sets designed to amplify fishes.

we combined the three 12S primer sets while combining the Riaz_12S and Berry_16S primers detected 93.4% (71 of 76). This finding supports the importance of not only employing multiple primer sets but also using primers that target different genes.

**Table 6. Results of SIMPER analysis conducted using the R package Vegan.**

| Taxon | Average dissimilarity | % contribution | % cumulative | Overall average dissimilarity |
|---|---|---|---|---|
| **Riaz_12S vs Berry_16S** | | | | 43.26 |
| *Mugil curema* | 15.04 | 34.76 | 34.76 | |
| *Leiostomus xanthurus* | 4.197 | 9.7 | 44.46 | |
| *Bairdiella chrysoura* | 3.723 | 8.606 | 53.07 | |
| *Pogonias cromis* | 3.026 | 6.994 | 60.06 | |
| *Lagodon rhomboides* | 2.735 | 6.321 | 66.38 | |
| **Riaz_12S vs MiFish_12S** | | | | 49.51 |
| *Mugil curema* | 17.67 | 35.68 | 35.68 | |
| *Bairdiella chrysoura* | 7.286 | 14.71 | 50.4 | |
| *Pogonias cromis* | 4.15 | 8.382 | 58.78 | |
| *Lagodon rhomboides* | 3.429 | 6.927 | 65.7 | |
| *Ariopsis felis* | 2.824 | 5.704 | 71.41 | |
| **Riaz_12S vs Valentini_12S** | | | | 43.87 |
| *Mugil curema* | 15.77 | 35.95 | 35.95 | |
| *Bairdiella chrysoura* | 6.508 | 14.84 | 50.78 | |
| *Lagodon rhomboides* | 3.632 | 8.281 | 59.06 | |
| *Pogonias cromis* | 3.394 | 7.738 | 66.8 | |
| *Ariopsis felis* | 2.364 | 5.389 | 72.19 | |
| **Berry_16S vs MiFish_12S** | | | | 45.68 |
| *Mugil curema* | 20.79 | 45.51 | 45.51 | |
| *Bairdiella chrysoura* | 5.267 | 11.53 | 57.04 | |
| *Leiostomus xanthurus* | 4.125 | 9.031 | 66.07 | |
| *Lagodon rhomboides* | 3.707 | 8.115 | 74.19 | |
| *Mugio cephalus* | 1.392 | 3.048 | 77.23 | |
| **Berry_16S vs Valentini_12S** | | | | 34.12 |
| *Mugil curema* | 12.77 | 37.41 | 37.41 | |
| *Bairdiella chrysoura* | 4.819 | 14.12 | 51.53 | |
| *Leiostomus xanthurus* | 3.497 | 10.25 | 61.78 | |
| *Lagodon rhomboides* | 3.185 | 9.335 | 71.12 | |
| *Lutjanus griseus* | 1.098 | 3.218 | 74.33 | |
| **MiFish_12S vs Valentini_12S** | | | | 43.05 |
| *Mugil curema* | 21.11 | 49.03 | 49.03 | |
| *Bairdiella chrysoura* | 8.486 | 19.71 | 68.74 | |
| *Lagodon rhomboides* | 6.283 | 14.59 | 83.33 | |
| *Mugil cephalus* | 1.528 | 3.549 | 86.88 | |
| *Leiostomus xanthurus* | 1.296 | 3.009 | 89.89 | |

The five species that contributed most to the dissimilarity among metabarcoding primer sets are listed.

The rate of evolution varies across genes and gene regions and therefore applying a single set of taxonomic thresholds (i.e., 99% sequence similarity for species-level designations) can be misleading. However, our analyses of the sequencing results from the four primer sets at three different species-level similarity thresholds (99%, 98%, and 97%) did not change our interpretation of the data. The Riaz_12S primer set either did as well or outperformed the others in terms of specificity (99.99% of reads assigned to fishes; Table 2), universality (greatest number of taxa amplified; S2 Table), and taxonomic resolution (96.78% of reads assigned to the species-level at the 99% threshold) regardless of which cutoff value was applied. Most notably, this

was true even when comparing Riaz_12S using the conservative 99% cutoff against all other primers sets using a 97% cut off (S2 Table). Similar patterns were found for diversity indices (S3 Table).

Despite the fact that the 18S primer set of MacDonald et al. [44] was designed to specifically target fish and a high annealing temperature (62˚C) was employed, less than 1% of the reads generated using these primers assigned to fishes and none could be assigned to the species level. Instead, >99% of reads were assigned to non-target taxa across a diversity of groups particularly copepods and green algae. This can likely be explained by the fact that these primers were designed based on the alignment of 18S gene sequences from only ten species of freshwater fish, and while they performed well in the original experiment where primer testing was performed on tissues from known fish species, they are not suitable for metabarcoding.

### 4.1 Detection of sharks and rays

All four primer sets tested here amplified DNA from both bony (class Actinopterygii) and cartilaginous fishes (class Chondrichthyes) with the vast majority of the reads (> 99%) assigned to the former (Table 4). Of the seven cartilaginous fish species detected, six were identified by Riaz_12S and four were detected by Valentini_12S. The MiFish_12S and Berry_16S detected only one species each. These results suggest that Riaz_12S primer set may also be suitable for the detection of sharks and rays. However, if elasmobranchs are the primary target group, it would be prudent to test primers using DNA extractions from species expected in the study area. Furthermore, there are a number of published primers that have been designed specifically for elasmobranchs that are worth exploring [12,60].

### 4.2 Reference databases and taxonomic assignments

Another factor that must be considered when choosing candidate primer sets for metabarcoding is the completeness of the reference databases [61,62]. For instance, while the COI reference databases for animals are robust, designing COI primers that are taxon specific is problematic, so primers are designed with high levels of degeneracy. This leads to the amplification of non-target taxa that can account for a large proportion of reads [28,29]. Furthermore, specific loci are often favored for some taxonomic groups. For example, 16S rRNA is most often employed to characterize bacterial communities, ITS is often used for fungi [63], and 18S rRNA is commonly employed for zooplankton [64]. For fishes, primers that amplify a portion of the 12S and 16S rRNA genes seem to provide a compromise between universality and specificity and are commonly used [65–67]. Incomplete reference databases also pose obstacles to taxonomic assignments [62,68]. Missing sequences can lead to misidentifications or the collapsing of assignments to genus or higher levels of classification. Over 20 years ago, when very few 16S rRNA bacterial sequences were publicly available, a 97% sequence similarity threshold was proposed for species-level assignments [69]. However, as publicly available sequence data increased exponentially, sequence similarity thresholds for species-level assignments have also increased and now typically range from 97–100% depending upon the target taxa and locus employed [27,35,70–74]. While there is no clear consensus on threshold criteria, recent studies have suggested that a 99–100% similarity threshold may be most appropriate for species-level assignments using 12S [73] and 16S rRNA markers [75].

### 4.3 Annealing temperature, human contamination, and other cautionary notes

Primer annealing temperature is an important factor in determining PCR success and specificity. At lower temperatures, just a partial match between primer and template can be sufficient

to permit amplification. On the other hand, higher annealing temperatures require exact or nearly exact primer-template match and usually results in high specificity. The MiFish_12S primer set has been employed in several published studies with annealing temperatures ranging from 50˚C to 65˚C [16,36,42,59,76–78]. Initially, we tested the MiFish_12S primer set using an annealing temperature of 55˚C based on Andruszkiewicz et al. [76]. Surprisingly, none of the resulting reads assigned to fish but instead were assigned to bacteria. Following the optimized protocol of Bylemans et al. [16], we raised the annealing temperature to 61.5˚C for subsequent experiments which resulted in 94.12% of reads assigning to fish. Our initial results could have been exacerbated by the high bacterial loads in our estuarine samples as DNA primers will preferentially amplify abundant templates and sometimes fail to amplify low abundant target DNA [17]. However, a similarly high percentage of non-target reads was observed by Miya et al. [79] for the MiFish_12S primers employed in marine waters where target DNA was also scarce. This example highlights the utility of testing primers on a small number of samples prior to purchasing large primer sets and/or the bulk processing of samples.

Human DNA was detected in all of our sequencing libraries except those produced using the Berry_16S primer set. Because human DNA was amplified in our field negative controls but not our lab controls (extraction and PCR negative controls), contamination was most likely introduced during sample collection or filtering. The presence of human DNA in eDNA metabarcoding studies is common [59,80,81]. To alleviate this problem, human-specific blocking primers have been used [82,83], however, the use of blocking oligos has been shown to reduce the number of target species detected in metabarcoding studies [42]. Moreover, only a small percentage (1.48%) of our reads were assigned to humans. Therefore, the use of blocking primers is not advised if contamination levels are likely to be low.

Many studies show a positive correlation between animal abundance and/or biomass estimates and the number of reads obtained from eDNA metabarcoding studies (reviewed by [84]). However, the inconsistency across studies, our insufficient understanding of DNA shedding rates, and the current paucity of information on how biotic and abiotic factors influence eDNA detection rates, limits the utility of eDNA for estimating species biomass and abundance [85,86]. These issues need to be more fully addressed before the relationship between eDNA copy number and abundance can be accurately modelled. As a result, we use read abundance as a proxy for species abundance for our Shannon's diversity value calculations but do so with great caution and refrain from over interpreting the results herein.

## 5 Conclusions and recommendations

The published data, including this study, demonstrate that for most aquatic systems no single primer set can capture all the diversity of any given community [10,87]. However, employing multiple primer sets may be cost prohibitive for some laboratories. In those cases where fish are the target the Riaz_12S primer set is a good option to consider. In our dataset from the biodiverse Indian River Lagoon in Florida, 99.9% of reads generated using this primer set assigned to fish and resulted in the greatest number of species detected. This is contrary to what seems to be a settling of some segments of the eDNA community on the MiFish_12S primers as the standard [88], which in our study did not perform as well. It should be noted that our results are based on limited sampling in a single, albeit highly diverse, estuary. Because fish communities vary significantly across space and time our findings may not be directly transferrable across systems. However, our results do add to the growing literature concerning eDNA protocols and best practices and will aid in the narrowing of primer choices for other researchers. Furthermore, our study highlights the importance of targeting different barcoding genes when possible. When we combined the results of the three 12S makers 85.5% species in the dataset

were accounted for, whereas 93.4% of the species were detected by combining the Riaz_12S and Berry_16S primer sets. The failings of the MacDonald_18S primer set to identify target taxa and the mis-priming of the MiFish_12S primer set due to low annealing temperature highlights the importance of methods testing and optimization before large investments are made in any particular protocol.

## Supporting information

**S1 Fig. Number of reads (log transformed) for each of the 17 fish species detected by all four primer sets (MiFish_12S, Riaz_12S, Valentini_12S, and Berry_16S).** Read numbers are totals across the six sample sites.
(TIF)

**S2 Fig. Nonmetric multidimensional scaling plots (NMDS) using presence/absence data for the fish species detected at six sampling locations in the Indian River Lagoon, Florida.** The numbers inside the circles represent sample sites. Metabarcoding data was generated using four primer sets designed to amplify fishes.
(TIF)

**S1 Table. Sampling sites used in this study. Two replicates of 500 ml water samples were collected in June 2018 from six locations within the Indian River Lagoon in central Florida.** Water samples were taken at the surface over shallow water habitat known to have oyster reefs.
(DOCX)

**S2 Table. Total number of fish species detected for each primer set at three different species-level sequence similarity thresholds (99%, 98%, and 97%).** Also shown is the percentage of total reads assigned at the species-level (fishes) after quality control.
(DOCX)

**S3 Table. Diversity indices including species richness, evenness, and Shannon's diversity calculated using the R package BiodiversityR v. 2.12–3 for four metabarcoding primer sets designed to amply fishes.** Standard deviations were calculated based on the data across all six sample sites. Data for three different species-level sequence similarity thresholds (99%, 98%, and 97%) are shown.
(DOCX)

**S4 Table. The number of reads obtained for each of the 17 fish taxa detected by all four primer sets used in this study.** Read numbers are totals across all the six sample sites.
(DOCX)

**S5 Table. Number of reads that were assigned to fish species at the 99% sequence similarity threshold for each primer set.** Total number of reads across all markers and the % of those total reads attributed to each marker is shown. Also listed is the average number of reads per species with the range in parenthesis.
(DOCX)

## Author Contributions

**Conceptualization:** Girish Kumar, Michelle R. Gaither.

**Data curation:** Girish Kumar, Ashley M. Reaume, Emily Farrell, Michelle R. Gaither.

**Formal analysis:** Girish Kumar.

**Funding acquisition:** Ashley M. Reaume, Emily Farrell, Michelle R. Gaither.

**Supervision:** Michelle R. Gaither.

**Writing – original draft:** Girish Kumar.

**Writing – review & editing:** Ashley M. Reaume, Emily Farrell, Michelle R. Gaither.

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
