## [Decision Letter · Decision Letter 0]

4 Aug 2021

PONE-D-21-15188

Choosing the best eDNA metabarcoding primer set for assessing fish communities in a biodiverse estuarine system

PLOS ONE

Dear Dr. Kumar,

Thank you for submitting your manuscript to PLOS ONE. After careful consideration, we feel that it has merit but does not fully meet PLOS ONE’s publication criteria as it currently stands. Therefore, we invite you to submit a revised version of the manuscript that addresses the points raised during the review process.

The manuscript represents an exploration of Fish diversity in an interesting ecosystem. The reviewers highlight several major issues that need to be addressed.

The first one may be with the angle of the writing, should this focus on exploring diversity or compare methods to explore the diversity. As the reviewers highlight, then this (in current form) falls short of doing a thorough comparison of the utility of the methods. E.g. rev.1 stresses that comparison of those methods should contrast – 1. specificity, 2. universality, and 3. resolution, but that the ms in current form only tackles 1. You may choose to add in silioco work and tackle 2 and 3, or reorient the paper towards using the different methods to tackle you study system(s) (and tone down conclusions of general lessons on methods).

The reviewers also call for beefing up of the analyses, most importantly dropping the assumption that read-covereage correlates with abundance (see rev. 3)

Also add information on sampling and background biology in paper and abstract (rev 2).

Finally, add intermediate datatables as supplemental data or cite open repositories where they can be accessed (data.dryad.org, etc).

We look forward to receiving your revised manuscript.

Kind regards,

Arnar Palsson, Ph.D.

Academic Editor

PLOS ONE

Journal Requirements:

Additional Editor Comments (if provided):

The manuscript represents an exploration of Fish diversity in an interesting ecosystem. The reviewers highlight several major issues that need to be addressed.

The first one may be with the angle of the writing, should this focus on exploring diversity or compare methods to explore the diversity. As the reviewers highlight, then this (in current form) falls short of doing a thorough comparison of the utility of the methods. E.g. rev.1 stresses that comparison of those methods should contrast – 1. specificity, 2. universality, and 3. resolution, but that the ms in current form only tackles 1. You may choose to add in silioco work and tackle 2 and 3, or reorient the paper towards using the different methods to tackle you study system(s) (and tone down conclusions of general lessons on methods).

The reviewers also call for beefing up of the analyses, most importantly dropping the assumption that read-covereage correlates with abundance (see rev. 3)

Also add information on sampling and background biology in paper and abstract (rev 2).

Finally, add intermediate datatables as supplemental data or cite open repositories where they can be accessed (data.dryad.org, etc).

Reviewers' comments:

Reviewer's Responses to Questions

**Comments to the Author**

1. Is the manuscript technically sound, and do the data support the conclusions?

Reviewer #1: No

Reviewer #2: Yes

Reviewer #3: Partly

2. Has the statistical analysis been performed appropriately and rigorously? 

Reviewer #1: No

Reviewer #2: Yes

Reviewer #3: No

3. Have the authors made all data underlying the findings in their manuscript fully available?

Reviewer #1: Yes

Reviewer #2: Yes

Reviewer #3: No

4. Is the manuscript presented in an intelligible fashion and written in standard English?

Reviewer #1: Yes

Reviewer #2: Yes

Reviewer #3: Yes

5. Review Comments to the Author

Reviewer #1: I have reviewed the manuscript by Kumar et al. entitled "Choosing the best eDNA metabarcoding primer set for assessing fish communities in a biodiverse estuarine system". The authors compared the usefulness of four primer sets for eDNA metabarcoding for fish at a model field of a lagoon in Florida. The four primer sites included three 12S (MiFish_12S, Valentini_12S, and Riaz_12S) and one 16S (Berry_16S) primers. The authors claimed that Riaz_12S and Berry_16S primer sets "detected" more numbers of species than MiFish_12S and Valentini_12S. Such comparison among metabarcoding primers is important for selection of assay systems, and in this sense, the manuscript has a merit to be published.

However, I have serious concerns on the methods for comparing primers. The usefulness of primers for eDNA metabarcoding should be evaluated the following three aspects, specificity (specific amplification of target taxa), universality (comprehensive amplification of target taxa without bias), and resolution (taxonomic resolution in the amplified region). The manuscript only evaluated primers based on the number of "detected" species. The criteria for species identification in this manuscript is ≧99% identity with sequences on the database, but this criteria is not fair. For example, the length of amplified region for Valentini_12S is short, and one base substitution make it genus-level identification according to the authors' criteria. Also, for MiFish_12S, many previous papers using this primer set adopt the identification threshold at ≧98.5% or lower, and from my inspection of these papers, I believe that the criteria of ≧99% identity is inappropriate. I recommend authors to set an appropriate threshold for each primer set, considering the difference of evolutionary speed among amplified regions. In addition, the accuracy of species determination may vary depending on the richness of the data in the database, so the results may vary greatly depending on the criteria set by the authors.

The above mentioned specificity, universality, and resolution can be tested via in silico test as well as using the data for field samples. Without such tests, the comparison among primer sets cannot be justified. From above reasons, I think the manuscript should be largely revised before its publication.

Reviewer #2: The manuscript submitted by Kumar et al. is a simple but well written piece focused on comparing different primer sets for fish eDNA metabarcoding. I would recommend the authors to improve the abstract and methods sections by including more necessary information, especially, regarding the sampling and studied area. For example, it is not very clear were exactly the samples were taken and if brackish or freshwater species are also expected to be recovered from those samples.

The abstract still needs to be modified by clarifying some important information (e.g., target taxonomic group, samples analysed, studied system, etc)

L27: Please include the referred ‘efforts’

L29: Include the target taxonomic group

L57: remove “a” in “a marine environments”

L87-89: The authors should better explore this information. The public databases might be more complete for some very few specific taxonomic groups. Therefore, when providing this information the target group might be included.

L114: Please include more information for the sampling sites

L166: How many runs?

L192-193: It is great to see the authors have adopted a conservative approach removing ASVs with query coverage below 100%.

L288: Insert a space between ‘and’’16S’

It is also important to note and discuss the lack of appropriate reference sequences. Despite being known that the combination of multiple primers targeting distinct gene is expected to increase taxonomic resolution and consequently, the number of species detected, the lack of references remains as a hindrance.

Reviewer #3: The authors performed a comparative study, aimed at evaluating the performances of different primer sets specifically designed to target fish species, in a highly biodiverse system: the Indian River Lagoon in Florida. They tested three 12S and one 16S rRNA primer sets, and a 18S primer set designed for freshwater fish that did not produce useful results. I found the manuscript clear, and I think its findings could be helpful for choosing the best primer sets for studying fish diversity using metabarcoding. However, I think that authors could make a bigger effort to present the differences among the tested primer sets in a more appropriate way. The figures also need to be supported by more precise legends.

Please, see below my comments:

Line 53-54: Here you cite some works aimed at describing environmental DNA metabarcoding in general. That is fine, except maybe for the paper of Kumar, Eble and Gaither (2020), that I find a little out of topic. I think that you should rather cite some empirical studies supporting your sentence. For example, the recent work of Aglieri et al. (2020) compared simultaneously three traditional sampling methods with environmental DNA metabarcoding, finding complementarity among the methods.

Line 98: duplicated “in”

Line 209: Except for species richness, the diversity indexes you used need abundance data. You used reads numbers as a proxy for abundance, but I am not so much a supporter of this approach. PCR biases can alter the proportion between the amount of initial DNA template and the final sequences yield. This is even more worthy of consideration when two PCRs are used to build the sequencing libraries, as you did. Moreover, the amount of genetic material in the water is not necessarily related to the abundance of individuals: different species have different dimensions, different shedding, etc. Several authors have made the attempt of relating sequence abundance to individual abundance, but the relation is very often poor. That is a current limit of metabarcoding, and at least you should explain that. Please, add a few lines to address this point.

Line 215: As in the previous comment. Bray-Curtis uses abundances. I would like to see also something made using presence/absence data, at least to verify the differences with the abundance related indexes you used. For example, you could show the same nMDS, but made with the Sørensen index. Please, show something more.

Line 216-217 and Figure 3: The nMDS plot shows six points for each primer set. I don’t get what they represent, since you sampled three sites taking four replicates for each one. The plot should show three points, in the case you pooled the replicates for each site, or 12 points if you were showing the diversity for each replicate. Please explain better what the points represent.

Line 222: Since you decided to consider the abundance of reads to calculate diversity indexes, why don’t you also show the number of reads of the taxa in common among the different primer sets? This could be indicative of the amplification efficiency of each primer set for those taxa. This is just a suggestion, but I would like to understand more.

Line 232: Please, could you provide also the average number of reads for sample?

Line 261-262: Low evenness means big differences in reads number among taxa. This is quite common in metabarcoding studies, since several variables can influence the yield, resulting in high variability. Anyway, how different these yields are for each taxon? Please, could you provide something more? For instance, a plot with all the reads of each taxon ordered from the smaller to the higher. Alternatives to my suggestion are also welcome.

6. PLOS authors have the option to publish the peer review history of their article (what does this mean?). If published, this will include your full peer review and any attached files.

Reviewer #1: No

Reviewer #2: No

Reviewer #3: No

---

## [Author Response · Author response to Decision Letter 0]

6 Oct 2021

Response to reviewer’s comments

Reviewer #1

Comment: I have serious concerns on the methods for comparing primers. The usefulness of primers for eDNA metabarcoding should be evaluated the following three aspects, specificity (specific amplification of target taxa), universality (comprehensive amplification of target taxa without bias), and resolution (taxonomic resolution in the amplified region). The manuscript only evaluated primers based on the number of "detected" species. 

Response: We agree that all three criteria are important and have clarified our goals in the introduction.

“Here we focus on the four best performing primer sets from the freshwater study of Zhang, Zhao (42) and include an additional 18S primer set not previously evaluated (but designed for fishes), and test these in a species rich estuarine system in Florida (the Indian River Lagoon). We compare the efficiency of these primers in terms of 1) specificity (amplification of fishes at the exclusion of other taxa), 2) universality (amplification of a diversity of fish taxa), and 3) taxonomic resolution (ability to resolve to the species level)”. 

By this definition we do address specificity. Our target group is fishes. In Table 2 we show the percent of reads that were off target (non-fish). For all primer sets at least 94% of reads assigned to fishes with the exception of the MacDonald primers. We also address universality in terms of the taxonomic levels that were amplified. We can’t directly address the question of sequencing bias with this dataset and therefore we do not attempt to translate read number into relative abundances, but we do include ASV files with read counts and their corresponding sequences. These files have been uploaded to Data Dryad (DOI: https://doi.org/10.5061/dryad.70rxwdbzc). In Table 3 we address the universality issue by highlighting the number of fish taxa (from class to species level) that were amplified, and Fig. 2 shows a Venn diagram of overlapping detections. 

Comment: The criteria for species identification in this manuscript is ≧99% identity with sequences on the database, but this criteria is not fair. For example, the length of amplified region for Valentini_12S is short, and one base substitution make it genus-level identification according to the authors' criteria. 

Response: We agree that some primer sets are better candidates for species delimitation. Our goal with this experiment was to determine which primer set allows us to identify the greatest number of fish species). Some gene segments will be better as they are more variable and able to delineate taxa more clearly. This is a function of length but also how the gene evolves and probably of more importance how complete the public databases are. Our goal was not to compare gene regions or consider fragment length, but instead we considered each of these primer sets to be a tool with its assigned task being to identify species in a water sample. Therefore, we evaluated each primer set as a published and established tool (including the public databases) and our goal was to determine which is the best for characterizing fish diversity in our estuarine system. 

Comment: Also, for MiFish_12S, many previous papers using this primer set adopt the identification threshold at ≧98.5% or lower, and from my inspection of these papers, I believe that the criteria of ≧99% identity is inappropriate. I recommend authors to set an appropriate threshold for each primer set, considering the difference of evolutionary speed among amplified regions. In addition, the accuracy of species determination may vary depending on the richness of the data in the database, so the results may vary greatly depending on the criteria set by the authors.

Response (L336-344): Regarding the threshold for the species identification, we used the following criteria for the taxonomic assignment: ≥ 99% similarity for species; 97% for genus; 95% for family; 90% for order; 85% for class; and 80% for phylum following West et al. 2020. The commonly used 97% sequence similarity threshold was proposed for species assignment in 1994 when only few 16S rRNA sequences were available in databases. With increase in the richness of sequences in databases over the years, a variety of sequence similarity threshold (97–100%) have been used in metabarcoding studies using different gene markers. Recent studies have suggested that a 99–100% similarity threshold may be more appropriate for species assignment using 12S (72) and 16S rRNA markers (74). Following these guidelines, we used ≥ 99% similarity cutoff for species assignment.

Comment: The above-mentioned specificity, universality, and resolution can be tested via in silico test as well as using the data for field samples. Without such tests, the comparison among primer sets cannot be justified. From above reasons, I think the manuscript should be largely revised before its publication.

Response (L98-104): In-silico testing of the primer sets used in our study (except McDonald_18S) was performed in Zhang et al. (2020). They found considerable discrepancies between the in-silico and in-vitro results including in the range and diversity of taxa amplified, fish community composition, and the power of each to discriminate species. Since the in-silico experiments had already been conducted (and the results were not always in agreement with in-vitro results), we went directly to in-vitro experiments testing the primers on field samples to evaluate their efficacy for fish biodiversity assessment. The explanation for this comment has been given in lines 98-104. 

Reviewer #2

Comment: The manuscript submitted by Kumar et al. is a simple but well written piece focused on comparing different primer sets for fish eDNA metabarcoding. I would recommend the authors to improve the abstract and methods sections by including more necessary information, especially, regarding the sampling and studied area. For example, it is not very clear were exactly the samples were taken and if brackish or freshwater species are also expected to be recovered from those samples.

The abstract still needs to be modified by clarifying some important information (e.g., target taxonomic group, samples analysed, studied system, etc)

Response: We thank the reviewer for pointing out these omissions. We have now included more details about our sample sites, target taxa, etc. in both the abstract and methods sections. Also, a table with GPS coordinates are now including in supplemental materials (Table S1). 

Comment: L27: Please include the referred ‘efforts’

Response (L26): Suggestion has been incorporated. Text now reads “The efficiency and outcome of these metabarcoding efforts are dependent upon…”

Comment: L29: Include the target taxonomic group

Response (L29): The target taxonomic group was fishes. The same has been added in line 29.

Comment: L57: remove “a” in “a marine environments”

Response (L60): Corrected

Comment: L87-89: The authors should better explore this information. The public databases might be more complete for some very few specific taxonomic groups. Therefore, when providing this information, the target group might be included.

Response (L88-89): We completely agree with the reviewer’s point of view. The markers used in this study are well represented for fishes in the public databases. The clarification has been given in line 89.

Comment: L114: Please include more information for the sampling sites.

Response: The details of sample information are now provided in the methods section and Table S1.

Comment: L166: How many runs?

Response (L176-178): The methods section now reads “Sequencing was conducted using a Nano 300 v2 (2 × 150) Reagent Kit for 2×111 cycles and a Nano 500 v2 (2 × 250) Reagent Kit for 2×251 cycles, based on amplicon size.”

Comment: L288: Insert a space between ‘and’’16S’

Response: Corrected

Comment: It is also important to note and discuss the lack of appropriate reference sequences. Despite being known that the combination of multiple primers targeting distinct gene is expected to increase taxonomic resolution and consequently, the number of species detected, the lack of references remains as a hindrance.

Response (L325-344): We completely agree that the effectiveness of eDNA metabarcoding studies are highly dependent on the completeness and accuracy of the relevant reference databases. We have discussed this issue in lines 325-344.

Reviewer #3: 

Comment: The authors performed a comparative study, aimed at evaluating the performances of different primer sets specifically designed to target fish species, in a highly biodiverse system: the Indian River Lagoon in Florida. They tested three 12S and one 16S rRNA primer sets, and a 18S primer set designed for freshwater fish that did not produce useful results. I found the manuscript clear, and I think its findings could be helpful for choosing the best primer sets for studying fish diversity using metabarcoding. However, I think that authors could make a bigger effort to present the differences among the tested primer sets in a more appropriate way. The figures also need to be supported by more precise legends.

Response: We thanks the reviewer for their thoughtful comments. We have improved the manuscript by adding more detailed information on sampling and plotting nMDS using both read abundance data as well as presence/absence data. The figure legends have been elaborated to include all the necessary information.

Comment: Line 53-54: Here you cite some works aimed at describing environmental DNA metabarcoding in general. That is fine, except maybe for the paper of Kumar, Eble and Gaither (2020), that I find a little out of topic. I think that you should rather cite some empirical studies supporting your sentence. For example, the recent work of Aglieri et al. (2020) compared simultaneously three traditional sampling methods with environmental DNA metabarcoding, finding complementarity among the methods.

Response: We have removed the review papers here and added a few empirical studies that better support this statement including Aglieri et al. (2020).

Comment: Line 98: duplicated “in”

Response: Corrected

Comment: Line 209: Except for species richness, the diversity indexes you used need abundance data. You used to read numbers as a proxy for abundance, but I am not so much a supporter of this approach. PCR biases can alter the proportion between the amount of initial DNA template and the final sequences yield. This is even more worthy of consideration when two PCRs are used to build the sequencing libraries, as you did. Moreover, the amount of genetic material in the water is not necessarily related to the abundance of individuals: different species have different dimensions, different shedding, etc. Several authors have made the attempt of relating sequence abundance to individual abundance, but the relation is very often poor. That is a current limit of metabarcoding, and at least you should explain that. Please, add a few lines to address this point.

Response (L345-353): We largely agree with the reviewer here. Even though several studies have shown positive correlations between abundance estimates from eDNA metabarcoding data (reviewed by Rourke et al. 2020), the correlation is often weak. We have address this comment in lines 345-353. 

Comment: Line 215: As in the previous comment. Bray-Curtis uses abundances. I would like to see also something made using presence/absence data, at least to verify the differences with the abundance related indexes you used. For example, you could show the same nMDS, but made with the Sørensen index. Please, show something more.

Response: As per the suggestion, we plotted the nMDS using abundance data as well as presence/absence data. Both the data set showed clear separation among primer sets. The nMDS plot using presence/absence data has been presented as a supplementary figure S2.

Comment: Line 216-217 and Figure 3: The nMDS plot shows six points for each primer set. I don’t get what they represent, since you sampled three sites taking four replicates for each one. The plot should show three points, in the case you pooled the replicates for each site, or 12 points if you were showing the diversity for each replicate. Please explain better what the points represent.

Response (L120-123): The six points in the nMDS plot represent the six sampling sites used in this study. From each site two field replicate water samples (500 ml per sample) were collected. The details of the sampling information has been provided in Materials and Methods section in lines 120-123 and Table S1.

Comment: Line 222: Since you decided to consider the abundance of reads to calculate diversity indexes, why don’t you also show the number of reads of the taxa in common among the different primer sets? This could be indicative of the amplification efficiency of each primer set for those taxa. This is just a suggestion, but I would like to understand more.

Response (L285-289): The number of reads of the taxa in common among the different primer sets have been provided in lines 285-289. The number of reads for the taxa in common among the Riaz_12S, Valentini_12S, and Berry_16S primer sets were comparable and 514,041, 555,883, and 569,976 reads were assigned to each primer set, respectively. The MiFish_12S has the lowest number of reads (381,805) assigned to species in common.

Comment: Line 232: Please, could you provide also the average number of reads for sample?

Response (L245-248): The average number of reads per sample for each marker set has been included in lines 245-248.

Comment: Line 261-262: Low evenness means big differences in reads number among taxa. This is quite common in metabarcoding studies, since several variables can influence the yield, resulting in high variability. Anyway, how different these yields are for each taxon? Please, could you provide something more? For instance, a plot with all the reads of each taxon ordered from the smaller to the higher. Alternatives to my suggestion are also welcome.

Response: As per the suggestion we plotted a graph of 17 common species detected across all four primer sets, with reads ordered from the smaller to the higher. The figure has been presented as a supplementary figure S1.

---

## [Decision Letter · Decision Letter 1]

4 Nov 2021

PONE-D-21-15188R1Choosing the best eDNA metabarcoding primer set for assessing fish communities in a biodiverse estuarine systemPLOS ONE

Dear Dr. Kumar,

Thank you for submitting your manuscript to PLOS ONE. After careful consideration, we feel that it has merit but does not fully meet PLOS ONE’s publication criteria as it currently stands. Therefore, we invite you to submit a revised version of the manuscript that addresses the points raised during the review process.

The manuscript has improved a great deal, but we opt for Major revision again because a major concern of reviewer 1 was not addressed (point 1). Also, because 2 reviewers backed out, I had to enlist two new ones, and one of them provided good recommendations that should improve the manuscript.

Change the threshold for species detection (>99%) is way to stringent. See rev 1. this AND previous comment! “I recommend authors to set an appropriate threshold for each primer set, considering the difference of evolutionary speed among amplified regions.”Because of the differences in amplicon length, then the comparison of the primer sets is challenging.Reviwer 5 offers suggestions on how to tackle this “I would suggest to use the proportions instead of the raw number of reads or at least transform the number of reads (e.g. forth square or logarithm) to make results more comparable. Comment: for a better comparison of the efficiency of the four different primer sets I would suggest the author to take into account not only the number of generated reads per marker but also their relative sequencing depth, especially for those sequences that were assigned to fishes”Provide more details on the sampling sites.Rev 5. “The authors state that they performed two replicates in each of the six sampling sites but then in the nMDS plot I can see only 6 points. Were the replicates pooled? How?... “Tone down the title (rev. “As a methodology study, this study is not rigorous enough to reflect the value of the title.”Also, the abstract ends with conflicting messages. You say that its best to use 2 or more primer sets, but then you conclude one set is better than another for your purposes?

We look forward to receiving your revised manuscript.

Kind regards,

Arnar Palsson, Ph.D.

Academic Editor

PLOS ONE

Additional Editor Comments:

The manuscript has improved a great deal, but we opt for Major revision again because a major concern of reviewer 1 was not addressed (point 1). Also, because 2 reviewers backed out, I had to enlist two new ones, and one of them provided good recommendations that should improve the manuscript.

1. Change the threshold for species detection (>99%) is way to stringent. See rev 1. this AND previous comment! “I recommend authors to set an appropriate threshold for each primer set, considering the difference of evolutionary speed among amplified regions.”

2. Because of the differences in amplicon length, then the comparison of the primer sets is challenging.

3. Reviwer 5 offers suggestions on how to tackle this “I would suggest to use the proportions instead of the raw number of reads or at least transform the number of reads (e.g. forth square or logarithm) to make results more comparable. Comment: for a better comparison of the efficiency of the four different primer sets I would suggest the author to take into account not only the number of generated reads per marker but also their relative sequencing depth, especially for those sequences that were assigned to fishes”

4. Provide more details on the sampling sites.

5. Rev 5. “The authors state that they performed two replicates in each of the six sampling sites but then in the nMDS plot I can see only 6 points. Were the replicates pooled? How?... “

6. Tone down the title (rev. “As a methodology study, this study is not rigorous enough to reflect the value of the title.”

7. Also, the abstract ends with conflicting messages. You say that its best to use 2 or more primer sets, but then you conclude one set is better than another for your purposes?

Reviewers' comments:

Reviewer's Responses to Questions

**Comments to the Author**

1. If the authors have adequately addressed your comments raised in a previous round of review and you feel that this manuscript is now acceptable for publication, you may indicate that here to bypass the “Comments to the Author” section, enter your conflict of interest statement in the “Confidential to Editor” section, and submit your "Accept" recommendation.

Reviewer #1: (No Response)

Reviewer #4: (No Response)

Reviewer #5: (No Response)

2. Is the manuscript technically sound, and do the data support the conclusions?

Reviewer #1: No

Reviewer #4: Partly

Reviewer #5: Partly

3. Has the statistical analysis been performed appropriately and rigorously? 

Reviewer #1: N/A

Reviewer #4: Yes

Reviewer #5: N/A

4. Have the authors made all data underlying the findings in their manuscript fully available?

Reviewer #1: Yes

Reviewer #4: Yes

Reviewer #5: Yes

5. Is the manuscript presented in an intelligible fashion and written in standard English?

Reviewer #1: Yes

Reviewer #4: Yes

Reviewer #5: Yes

6. Review Comments to the Author

Reviewer #1: The authors kept ≥ 99% similarity cutoff for species assignment regardless of the previous review comments. The threshold should be set for each primer set because the evolutionary rate varies even within the same gene. From my experience for example, The MiFish region clearly has a fast evolutionary rate, and one underestimates the species detection when a 99% threshold is adopted. The threshold value should be set carefully because it has a significant impact on the validity of primer selection.

Reviewer #4: In my opinion, the paper is mainly sound. However, the authors clarified the candidate primers were chosen according to Zhang (2021). Zhang (2021) is based on freshwater fish and suggested that the primer choice cannot be solely based on in silico evaluation. As the author said by themselves, primer selection experiments should do both in silico and vitro. So I think that's not enough to justify the choice of these candidate primers. The author should clarify this point in the discussion because marine samples are very different from freshwater samples. Otherwise, this study doesn't have a customer database to compare the outcome species, but it's fine as it's fair to all primers. They only use the inventory list alone is not enough to say that they did an excellent species resolution survey. In short, The authors show how they selected primers for their study in an estuarine environment. It's a good complement to the eDNA study. As a methodology study, this study is not rigorous enough to reflect the value of the title. The authors already account for the shortcomings of their experimental design in the revised edition. I suggest authors should also clarify the limitations of candidate primer selection.

Reviewer #5: Dear Editor,

I find myself in agreement with reviewer #1 and I think the manuscript still needs substantial revision before its publication, and also feel that reviewer 1 should have the opportunity to assess the authors’ responses.

I add some more personal comments to the previous reviewer’s ones.

Comment: I generally have serious concerns regarding the idea of comparing primers that have such a different amplicon length and especially I agree with reviewer #1 in the criticism about the criteria for species identification, using the same threshold for example for Valentini 12S that is 63 bp long and for MiFish 12S, which is nearly three times that length.

Comment: the authors used reads numbers as a proxy for abundance data and compared the results for the four primer sets. A lot of factors can affect the proportion between the initial amount of template DNA and the final numbers of sequence reads. This is even more an issue considering the different lengths of the fragment amplified and the fact that you used two PCRs to build sequencing libraries. I would suggest to use the proportions instead of the raw number of reads or at least transform the number of reads (e.g. forth square or logarithm) to make results more comparable.

Comment: for a better comparison of the efficiency of the four different primer sets I would suggest the author to take into account not only the number of generated reads per marker but also their relative sequencing depth, especially for those sequences that were assigned to fishes.

Comment: The authors state that they performed two replicates in each of the six sampling sites but then in the nMDS plot I can see only 6 points. Were the replicates pooled? How? I think this should be specified, as it will also affect species richness values within samples.

Comment: I agree with reviewer #2 and I think you should add more details on the sampling sites. I would suggest to add a map showing the geographic coordinates of each sampling site or at least add more details in Table S1. Latitude and longitude records should obviously include N/S and W/E annotations respectively.

7. PLOS authors have the option to publish the peer review history of their article (what does this mean?). If published, this will include your full peer review and any attached files.

Reviewer #1: No

Reviewer #4: No

Reviewer #5: No

---

## [Author Response · Author response to Decision Letter 1]

3 Feb 2022

Response to reviewer’s comments

Reviewer #1: 

Comment: The authors kept ≥ 99% similarity cutoff for species assignment regardless of the previous review comments. The threshold should be set for each primer set because the evolutionary rate varies even within the same gene. From my experience for example, The MiFish region clearly has a fast evolutionary rate, and one underestimates the species detection when a 99% threshold is adopted. The threshold value should be set carefully because it has a significant impact on the validity of primer selection.

Response: We agree. As per the suggestion of the reviewer, we used three different similarity cutoffs (97%, 98%, and 99%) to see if the change in threshold criteria influenced species detection. An increase in the total number of detected species was observed when we used 98% cutoff as compared to 99% similarity threshold (Table S2; below). However, a comparison of 98% to 97% similarity threshold did not lead to a change (except one species for MiFish). Similar to taxonomic assignment, different cutoffs for the primers did not change the overall diversity indices (Table S3). Since the overall result did not change, we decided to use 99% similarity threshold to avoid any misidentification of species. 

Table S2 Fish species detected for each primer set at each of three different similarity thresholds (99%, 98%, and 97%). Also shown is the percentage of total reads assigned at the species-level (fishes) after quality control.

Primer set 99% Similarity threshold 98% Similarity threshold 97% Similarity threshold

 # Species % Reads assigned to species # Species % Reads assigned to species # Species % Reads assigned to species

MiFish_12S 34 81.39 40 84.71 41 84.91

Riaz_12S 54 96.78 61 97.65 61 97.65

Valentini_12S 32 86.03 37 87.43 37 87.43

Berry_16S 48 97.38 51 97.86 51 97.88

We’ve added this text to section 3.2

“While adjusting the sequence similarity threshold did change the number of species detected for each primer set the overall pattern did not change. Regardless of which similarity threshold was applied the Riaz_12S primer set resulted in the most species detected with Berry-16S ranking second (Table S2). There were no notable changes in taxa detected when comparing the 97% and 98% thresholds but there was an increase in the number of species resolved if we applied the 98% threshold compared to the more conservative 99% threshold. Most notably, when we compared the best performing Riaz_12S using the 99% cutoff with the worst performing primers sets using a 97% cut off the latter still resulted in high number of specie detections (Table S2). Similar patterns were found for diversity indices (Table S3). As a result, we have opted to apply a single conservative similarity threshold of 99% for species-level designations and report these throughout.”

Reviewer #4: In my opinion, the paper is mainly sound. However, the authors clarified the candidate primers were chosen according to Zhang (2021). Zhang (2021) is based on freshwater fish and suggested that the primer choice cannot be solely based on in silico evaluation. As the author said by themselves, primer selection experiments should do both in silico and vitro. So I think that's not enough to justify the choice of these candidate primers. The author should clarify this point in the discussion because marine samples are very different from freshwater samples. Otherwise, this study doesn't have a customer database to compare the outcome species, but it's fine as it's fair to all primers. They only use the inventory list alone is not enough to say that they did an excellent species resolution survey. In short, the authors show how they selected primers for their study in an estuarine environment. It's a good complement to the eDNA study. As a methodology study, this study is not rigorous enough to reflect the value of the title. The authors already account for the shortcomings of their experimental design in the revised edition. I suggest authors should also clarify the limitations of candidate primer selection.

Response: The choice of primer candidates has been further explained in the discussion section. Zhang et al. (2020) performed in silico experiments on a number of primer sets including 4 of the 5 selected here. For in vitro experiments they tested these same primer sets in a freshwater system. The title of the manuscript has been changed to reflect our study.

Our Discussion now reads

“Despite the growing interest in the use of eDNA to assess fish communities, there has been a surprising lack of studies that have directly compared the efficacy of metabarcoding primer sets in marine and estuarine systems. Because the performance of eDNA metabarcoding primers will vary depending on the study system (freshwater, marine, or estuarine) and taxonomic composition, there is no guarantee that a primer set that performs well in a freshwater system will do so in marine or estuarine systems. For this reason, we tested the four best performing metabarcoding primers from the freshwater study of Zhang, Zaho (42), to determine their effectiveness in a biodiverse estuarine system.”

Reviewer #5: 

Comment: I generally have serious concerns regarding the idea of comparing primers that have such a different amplicon length and especially I agree with reviewer #1 in the criticism about the criteria for species identification, using the same threshold for example for Valentini 12S that is 63 bp long and for MiFish 12S, which is nearly three times that length.

Response: The reviewers point out the important point that the primers used in this study produce different amplicon lengths which may influence species detection. It is also important to take into consideration that when a lab picks up a new protocol, they are looking for clear guidance concerning thresholds and cuts-off but in general this is lacking in the literature. As a result, general rules of thumb are typically applied such as 99% for species level designations. With this in mind we now present some analyses using three different similarity cutoffs (97%, 98%, and 99%) to determine if changes in threshold criteria influence the total number of species detected. Overall interpretation of the results did not change, and our results section (Section 3.2) now reads:

“While adjusting the sequence similarity threshold did change the number of species detected for each primer set the overall pattern did not change. Regardless of which similarity threshold was applied the Riaz_12S primer set resulted in the most species detected with Berry-16S ranking second (Table S2). There were no notable changes in taxa detected when comparing the 97% and 98% thresholds but there was an increase in the number of species resolved if we applied the 98% threshold compared to the more conservative 99% threshold. Most notably, when we compared the best performing Riaz_12S using the 99% cutoff with the worst performing primers sets using a 97% cut off the latter still resulted in high number of specie detections (Table S2). Similar patterns were found for diversity indices (Table S3). As a result, we have opted to apply a single conservative similarity threshold of 99% for species-level designations and report these throughout.”

Furthermore, our discussion now reads

“Because the rate of evolution varies across genes and gene regions applying a single set of taxonomic thresholds (i.e., 99% for species; 97% for genus; 95% for family; 90% for order; 85% for class; and 80% for phylum; as applied here) can be misleading. However, our analyses of the sequencing results from the four primer sets at three different species-level similarity thresholds (99%, 98%, and 97%) did not change our interpretation of the data. The Riaz_12S primer set either did as well or outperformed the others in terms of specificity (99.99% of reads assigned to fishes; Table 2), universality (greatest number of taxa amplified; Table S2), and taxonomic resolution (96.78% of reads assigned to the species-level at the 99% threshold) regardless of which cutoff value was applied. Most notably, this was true even when comparing Riaz_12S using the conservative 99% cutoff against all other primers sets using a 97% cut off (Table S2). Similar patterns were found for diversity indices (Table S3). It’s important to note that the Berry_16S primer set also performed well in our experiments.”

Comment: the authors used reads numbers as a proxy for abundance data and compared the results for the four primer sets. A lot of factors can affect the proportion between the initial amount of template DNA and the final numbers of sequence reads. This is even more an issue considering the different lengths of the fragment amplified and the fact that you used two PCRs to build sequencing libraries. I would suggest to use the proportions instead of the raw number of reads or at least transform the number of reads (e.g. forth square or logarithm) to make results more comparable.

Response: We thank the reviewer for this suggestion. We have log transformed the sequence reads of the 17 common species among all the four primer sets used in this study and a figure presenting these data can now be found as Figure S2.

Our text now reads:

“Based on the ANOSIM analyses, species assemblages differed significantly across primer sets (R = 0.4011, P < 0.001); a finding that was supported by the NMDS plots which showed clear separation among marker sets when either the read abundance data (Figure 3) or presence/absence data (Figure S2) were analyzed.”

Comment: for a better comparison of the efficiency of the four different primer sets I would suggest the author to take into account not only the number of generated reads per marker but also their relative sequencing depth, especially for those sequences that were assigned to fishes.

Response: Percent of total reads per marker and average sequencing depth per species were calculated and data is now presented in Table S5. Our results were further corroborated as MiFish_12S resulted in the number of reads suggesting lower PCR efficiency for this primer set in our experiments.

Comment: The authors state that they performed two replicates in each of the six sampling sites but then in the nMDS plot I can see only 6 points. Were the replicates pooled? How? I think this should be specified, as it will also affect species richness values within samples.

Response: The sequence reads of replicates from the same sites were pooled to plot NMDS. This has been clarified in the text in line 230..

Comment: I agree with reviewer #2 and I think you should add more details on the sampling sites. I would suggest to add a map showing the geographic coordinates of each sampling site or at least add more details in Table S1. Latitude and longitude records should obviously include N/S and W/E annotations respectively.

Response: The details of our study sites has been added to Table S1 with GPS coordinates in decimal degrees (DD) which is the preferred format for GEOME where we archive our metadata. DD are not typically reported with N/S or W/E annotations.

---

## [Decision Letter · Decision Letter 2]

8 Mar 2022

PONE-D-21-15188R2Comparing eDNA metabarcoding primers for assessing fish communities in a biodiverse estuaryPLOS ONE

Dear Dr. Kumar,

Thank you for submitting your manuscript to PLOS ONE. After careful consideration, we feel that it has merit but does not fully meet PLOS ONE’s publication criteria as it currently stands. Therefore, we invite you to submit a revised version of the manuscript that addresses the points raised during the review process.

The manuscript is in generally good shape. I ended with minor revision, one recommendation and a list of small suggestions.

The study compared the primer pairs, but did not make an attempt to describe the differences in biological resolution. Do all primer pairs have the same power to capture differences between locations (or any other biological variable of interest, habitats, seasons, depth etc)? It is not described in the methods (or I missed it), but do the 6 sites differ biologically?

This can be tackled easily, can you add location numbers to NMDS graph. Also, can you test for differences between sites?

Please bring this point also up in the discussion.

We look forward to receiving your revised manuscript.

Kind regards,

Arnar Palsson, Ph.D.

Academic Editor

PLOS ONE

Journal Requirements:

Additional Editor Comments (if provided):

PONE-D-21-15188R2

"Comparing eDNA metabarcoding primers for assessing fish communities in a biodiverse estuary"

The manuscript is in generally good shape. I ended with minor revision, one recommendation and a list of small suggestions.

The study compared the primer pairs, but did not make an attempt to describe the differences in biological resolution. Do all primer pairs have the same power to capture differences between locations (or any other biological variable of interest, habitats, seasons, depth etc)? It is not described in the methods (or I missed it), but do the 6 sites differ biologically?

This can be tackled easily, can you add location numbers to NMDS graph. Also, can you test for differences between sites?

Please bring this point also up in the discussion.

Minor points.

Line 64

“breadth of which can vary from”

Line 98.

References to the Zhang paper, no need to include the first name, “Zhang et al” is better

“In a recent study, Zhang et al. (42)”

Line 209. Extra word in citation?

“following West, Stat (35).”

Line 230-31. Another refernce issue”

“An analysis of similarity (ANOSIM; ANOSIM; 55)”

Line 272. Add “predictably”

“but there was predictably an increase in the number”

Line 321.

Is there any guarantee that the primers found here will be best in other marine or estuarine systems?

Line 350. Ref issues again, sort through entire manuscript and fix.

“MacDonald, Young (44)”

Line 365 Please rephrase.

“However, careful testing is still needed.”

Reviewers' comments:

Reviewer's Responses to Questions

**Comments to the Author**

1. If the authors have adequately addressed your comments raised in a previous round of review and you feel that this manuscript is now acceptable for publication, you may indicate that here to bypass the “Comments to the Author” section, enter your conflict of interest statement in the “Confidential to Editor” section, and submit your "Accept" recommendation.

Reviewer #1: All comments have been addressed

2. Is the manuscript technically sound, and do the data support the conclusions?

Reviewer #1: Yes

3. Has the statistical analysis been performed appropriately and rigorously? 

Reviewer #1: Yes

4. Have the authors made all data underlying the findings in their manuscript fully available?

Reviewer #1: Yes

5. Is the manuscript presented in an intelligible fashion and written in standard English?

Reviewer #1: Yes

6. Review Comments to the Author

Reviewer #1: (No Response)

7. PLOS authors have the option to publish the peer review history of their article (what does this mean?). If published, this will include your full peer review and any attached files.

Reviewer #1: No

---

## [Author Response · Author response to Decision Letter 2]

24 Mar 2022

Comment: The study compared the primer pairs but did not make an attempt to describe the differences in biological resolution. Do all primer pairs have the same power to capture differences between locations (or any other biological variable of interest, habitats, seasons, depth etc.)? It is not described in the methods (or I missed it) but do the 6 sites differ biologically?

This can be tackled easily; can you add location numbers to NMDS graph. Also, can you test for differences between sites?

Please bring this point also up in the discussion.

Response: Thank you for the positive feedback on our manuscript. Our sample sites don’t represent different habitats but instead are over a similar bottom type and are in close proximity. Therefore, we consider them to be replicates. However, that still begs the question of whether they are different. So, as you suggested, we performed the ANOVA to determine if the sites differ. Results showed no significant differences among the sites (p = 0.33), suggesting that the sampling sites are biologically similar. In the NMDS plot (figure 3), we have added the location numbers and clarified in the legend of figure. All these points have been clarified in the methods/results and mentioned in the discussion section.

Minor points.

Line 64

“breadth of which can vary from”

Response: Corrected

Line 98.

References to the Zhang paper, no need to include the first name, “Zhang et al” is better

“In a recent study, Zhang et al. (42)”

Response: Done

Line 209. Extra word in citation?

“following West, Stat (35).”

Response: Citation has been corrected.

Line 230-31. Another reference issue”

“An analysis of similarity (ANOSIM; ANOSIM; 55)”

Response: Duplicate ANOSIM has been deleted.

Line 272. Add “predictably”

“but there was predictably an increase in the number”

Response: “Predictably” has been added in the sentence.

Line 321.

Is there any guarantee that the primers found here will be best in other marine or estuarine systems?

Response: There is no guarantee that performance of primers in this study will be best in other marine or estuarine system as effectiveness of primers differ depending on the geographic regions and different ecosystems. The same has been discussed in the Conclusions and Recommendations section.

Line 350. Ref issues again, sort through entire manuscript and fix.

“MacDonald, Young (44)”

Response: Corrected

Line 365 Please rephrase.

“However, careful testing is still needed.”

Response: The section now reads “However, if elasmobranchs are the primary target group, it would be prudent to test primers using DNA extractions from species expected in the study area. Furthermore, there are a number of published primers that have been designed specifically for elasmobranchs that are worth exploring”

---

## [Editor Report · Decision Letter 3]

28 Mar 2022

Comparing eDNA metabarcoding primers for assessing fish communities in a biodiverse estuary

PONE-D-21-15188R3

Dear Dr. Kumar,

We’re pleased to inform you that your manuscript has been judged scientifically suitable for publication and will be formally accepted for publication once it meets all outstanding technical requirements.

Kind regards,

Arnar Palsson, Ph.D.

Academic Editor

PLOS ONE
---

## [Editor Report · Acceptance letter]

1 Apr 2022

PONE-D-21-15188R3 

Comparing eDNA metabarcoding primers for assessing fish communities in a biodiverse estuary 

Dear Dr. Kumar:

I'm pleased to inform you that your manuscript has been deemed suitable for publication in PLOS ONE. Congratulations! Your manuscript is now with our production department. 

Kind regards, 

on behalf of

Dr. Arnar Palsson 

Academic Editor

PLOS ONE